

# Extreme weather exposure identification for road networks – a comparative assessment of statistical methods

Matthias Schlögl[1], Gregor Laaha[2]

[1]Mobility Department – Transportation Infrastructure Technologies, Austrian Institute of Technology, Vienna, 1210, Austria
[2]Institute of Applied Statistics and Scientific Computing, University of Natural Resources and Life Sciences, Vienna, 1190, Austria

*Correspondence to*: Matthias Schlögl (matthias.schloegl.fl@ait.ac.at)

**Abstract.** The assessment of road infrastructure exposure to extreme weather events is of major importance for scientists and practitioners alike. In this study, we compare the different extreme value approaches and fitting methods with respect to their value for assessing the exposure of transport networks to extreme precipitation and temperature impacts. Based on an Austrian data set from 25 meteorological stations representing diverse meteorological conditions, we assess the added value of partial duration series over the standardly used annual maxima series in order to give recommendations for performing extreme value statistics of meteorological hazards. Results show the merits of the robust L-moment estimation, which yielded better results than maximum likelihood estimation in 62 % of all cases. At the same time, results question the general assumption of the threshold excess approach (employing partial duration series, PDS) being superior to the block maxima approach (employing annual maxima series, AMS) due to information gain. For low return periods (non-extreme events) the PDS approach tends to overestimate return levels as compared to the AMS approach, whereas an opposite behaviour was found for high return levels (extreme events). In extreme cases, an inappropriate threshold was shown to lead to considerable biases that may outperform the possible gain of information from including additional extreme events by far. This effect was neither visible from the square-root criterion, nor from standardly used graphical diagnosis (mean residual life plot), but from a direct comparison of AMS and PDS in synoptic quantile plots. We therefore recommend performing AMS and PDS approaches simultaneously in order to select the best suited approach. This will make the analyses more robust, in cases where threshold selection and dependency introduces biases to the PDS approach, but also in cases where the AMS contains non-extreme events that may introduce similar biases. For assessing the performance of extreme events we recommend conditional performance measures that focus on rare events only in addition to standardly used unconditional indicators. The findings of the study directly address road and traffic management, but can be transferred to a range of other environmental variables including meteorological and hydrological quantities.

## 1 Introduction

Reliable information about the exposure of road infrastructure networks to extreme weather events is of major concern for road authorities, governmental institutions and safety researchers all over the world (TRB, 2008; Koetse and Rietveld, 2009;





Eisenack et al., 2011; Doll et al., 2013; UNECE, 2013; Meyer et al., 2014; Michaelides, 2014; Schweikert et al., 2014a, 2014b; Matulla et al., 2016). In a changing climate (IPCC, 2012) and due to extensive soil sealing (Nestroy, 2006) the impact of extreme weather events are likely to increase in both frequency and intensity (APCC, 2014). Against this background, the resilience of transport systems with respect to weather hazards has become increasingly important.

A basic requirement for foresightful road infrastructure management are data about both the probability and magnitude of severe weather events. This information can be derived from long-term records of weather quantities such as precipitation and temperature, by means of statistical extreme value modeling. While extreme value theory provides a methodological framework that is commonly used in various scientific disciplines, such as hydrology (Katz et al., 2002), finance (Embrechts et al., 2003), engineering (Castillo et al., 2005) and climate sciences (Katz, 2010; Cheng et al., 2014), the application of these

tools for road network exposure analysis is a relatively uncharted area. In particular, formal comparative assessments of the various statistical methods that can be applied for estimating return levels of extreme events are rare.

Two basic approaches have been proposed for deriving extreme value series (Coles, 2001), which are both widely applied in studying extreme meteorological events (e.g. Smith, 1989; Davison and Smith, 1990; Parey et al., 2010; Villarini, 2011; Papalexiou and Koutsoyiannis, 2013). On the one hand, the maximum value per year can be used in the block maxima

approach, resulting in an annual maxima series (AMS). On the other hand, all values exceeding a certain threshold can be considered extreme, leading to the threshold excess approach based on partial duration series (PDS). Once the extreme value series has been derived, an appropriate distribution function is fitted to these observations by using different parameter estimation methods, such as maximum likelihood estimation, method of moments or Bayesian methods for parameter estimation. Clearly, there are a number of possible combinations of the approaches that may lead to different, often equally

plausible results.

Several efforts have been made to compare the performance of block maxima and threshold excess approaches. While some studies only provide a qualitative description of resulting parameter estimates and estimated return levels for both methods (Jarušková and Hanek, 2006), more formal assessment approaches are based on the asymptotic variance of the T-year event estimator (Cunnane, 1973) or on various goodness-of-fit tests and model performance metrics (Madsen et al., 1997a, 1997b;

Bezak et al., 2014). Controversial conclusions have been drawn. For instance, Madsen et al. (1997a) found for extreme discharges that the most suitable approach depends on the sample size and the shape parameter of the fitted functions. However, Ben-Zvi (2009) and Bezak et al. (2014) argue that a Generalized Pareto distribution fitted to partial duration series yields the best results for modelling rainfall and discharge extremes. Mkhandi et al. (2005), again, found that AMS and PDS methods result in similar predictions of flood magnitudes. All of these studies document the importance of extreme value

analysis in hydrology, but we are not aware of similar studies on temperature extremes that are equally important as rainfall impacts for road networks. Moreover, the studies did not specifically assess the performance of methods with respect to rare events, such as 100-year events, which are more relevant for risk assessments than events at the moderate tail of the distribution.



In this study, we compare the different extreme value approaches and fitting methods with respect to their value for assessing the exposure of transport networks to extreme weather impacts. Based on an Austrian data set from 25 meteorological stations representing diverse meteorological conditions, we assess the added value of partial duration series over the standardly used annual maxima series in order to give recommendations for performing extreme value statistics of
meteorological hazards.

## 2 Materials and methods

### 2.1 Data – Meteorological indicators

This study focuses on several meteorological indicators that can be used to assess the exposure of road networks to two main meteorological quantities: precipitation and temperature. These two variables are considered to have the most serious
influence on damage to infrastructure (Matulla et al., in press). They are measured by meteorological services on a regular basis so the data quality is usually high. Nevertheless, the methodology presented in this paper is applicable to various other meteorological quantities (e.g. maximum wind speed), if time-series of about 30 years or more are available.

Four meteorological indices are used in this study. Temperature impacts are considered by daily minimum ($T_{min}$) and daily maximum temperature ($T_{max}$). In addition, maximum daily temperature difference ($T_\Delta = T_{max} - T_{min}$) is analysed, with all
temperature indices in [°C]. Regarding precipitation impacts, the daily precipitation sum [mm/d] has been chosen.

In order to identify suitable meteorological stations that represent the main climate features of the highway network in Austria, all monitoring stations operated by the national weather service Zentralanstalt für Meteorologie und Geodynamik (ZAMG) served as a starting point. The selection of suitable stations was carried out in a stepwise procedure with respect to the following considerations: Firstly, the spatial proximity of available measuring stations to the highway network was
considered, by excluding stations with a distance greater than 10 kilometres from the data set. Secondly, data availability and data quality were considered. As sufficiently long time series are a prerequisite for reliable return level estimation, only stations with more than 30 years of record (i.e., since 1 January 1985) and with less than 5% missing values were selected. Finally, topographic conditions and regional peculiarities were taken into account for selecting evenly spread and climatically representative stations. This step was guided by visual inspection of climate maps (Hiebl et al., 2011) and the
digital hydrological atlas of Austria (BMLFUW, 2007; Fürst et al., 2009). The dataset so obtained consists of 25 hot spots representing climatically homogeneous regions of Austria (Fig. 1).



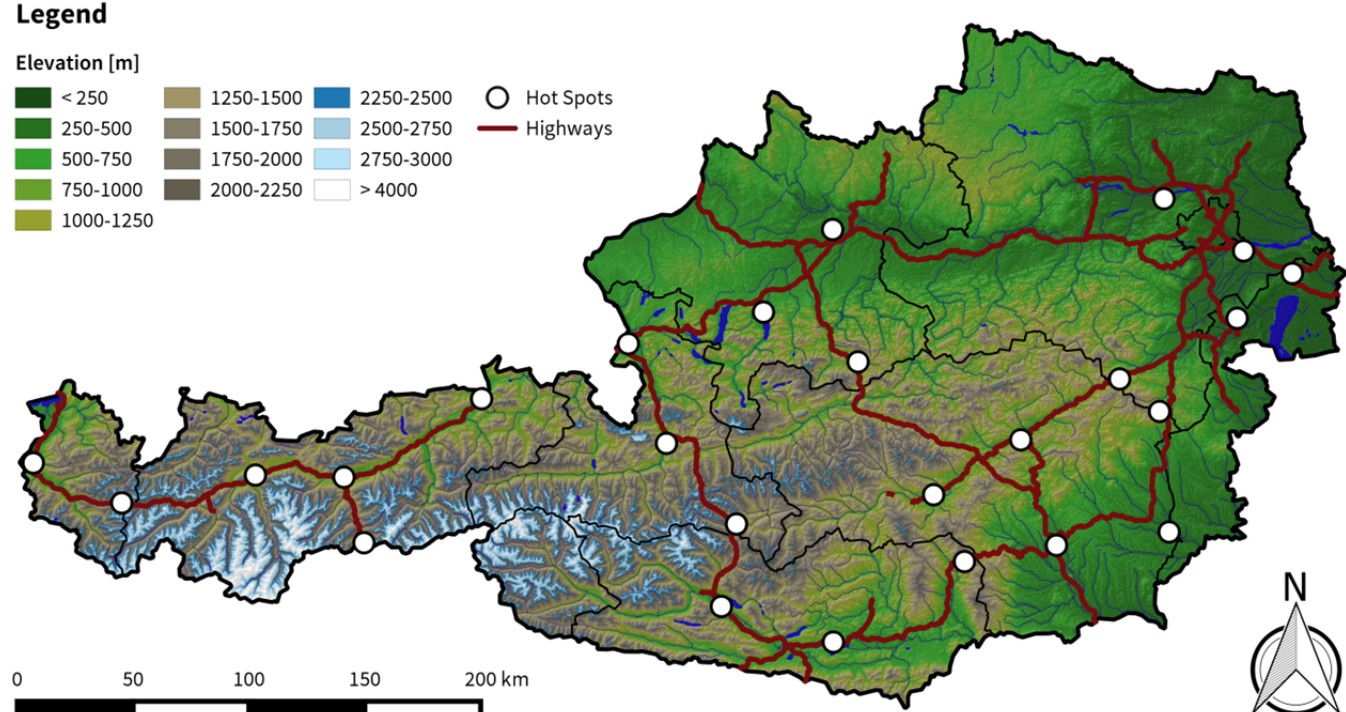

**Figure 1: Location of the selected meteorological stations used for extreme value analysis.**

## 2.2 Extreme value selection

### 2.2.1 Block maxima method

The first approach for deriving extreme value series consists in selecting maximum (or similarly minimum) values of the observations within subsequent time intervals (blocks) of constant length. While the block size is freely selectable, a trade-off has to be made between bias (small blocks) and variance (large blocks). Most commonly, the length of the block is chosen to correspond to a calendar year (Coles, 2001), resulting in an annual extreme value series. This was also the case in our study.

Based on the Fisher–Tippett–Gnedenko theorem, a generalized extreme value (GEV) distribution is appropriate for modelling the resulting annual maxima series (Fisher and Tippet, 1928; Gnedenko, 1943). The cumulative distribution function of the GEV is defined by

$$G_{\mu,\sigma,\xi}(z) = exp\left\{-\left[1 + \xi\left(\frac{z-\mu}{\sigma}\right)\right]^{-1/\xi}\right\} \tag{1}$$

for the set $\left\{z: 1 + \xi\left(\frac{z-\mu}{\sigma}\right) > 0\right\}$ where $\mu$ is the location parameter, $\sigma$ is the scale parameter and $\xi$ is the shape parameter. Alternative formulations with inverse sign of $\xi$ are also common (e.g. Hosking, 1990). In both cases, the parameters satisfy

$-\infty < \mu < \infty$, $\sigma > 0$ and $-\infty < \xi < \infty$ (Coles, 2001).





The GEV comprises three different types of distributions, which can be distinguished by the sign of their shape parameter: Gumbel, Fréchet and Weibull distribution (Fréchet, 1927; Gumbel, 1958; Coles, 2001; Embrechts et al., 2003; Basrak, 2014). The Gumbel distribution is commonly applied for maxima that are not limited towards an upper bound, whereas the Weibull case is more appropriate for minima which are often limited by a lower bound (Tallaksen and van Lanen, 2004).

### 2.2.2 Threshold excess method

In some cases, fitting distributions to block maxima data is a wasteful approach as only one value per block is used for modelling. A threshold excess approach potentially provides more information on extremes (Coles, 2001).

Analogous to the choice of the block size in the block maxima approach, the selection of the threshold value in the threshold excess method is also subject to a trade-off between bias (due to selecting non-extreme events if the threshold is low) and variance (due to a small number of exceedances when selecting a high threshold). Hence, the choice of a suitable threshold is important. The basic aim is to select the potentially lowest threshold, given the prerequisite that the extreme value model must provide a reasonable approximation to exceedances above this threshold and shall not contain non-extreme events (Coles, 2001). According to the Pickands–Balkema–de Haan theorem, a Generalized Pareto (GP) distribution is suited for modelling the resulting threshold excesses (Balkema and de Haan, 1974; Pickands, 1975): It states that, for some large threshold $u$, the distribution function of $(X - u)$, conditional on $X > u$ can be well approximated by the Generalized Pareto distribution, which is defined by

$$H_{\xi,\sigma}(z) = \begin{cases} 1 - \left[1 + \xi\left(\frac{z-\mu}{\sigma}\right)\right]^{-1/\xi} & \text{for } \xi \neq 0 \\ 1 - exp\left(-\frac{z-\mu}{\sigma}\right) & \text{for } \xi = 0 \end{cases} \tag{2}$$

where the support is $z \geq \mu$ in the case $\xi \geq 0$, and $\mu \leq z \leq \mu - \sigma/\xi$ when $\xi < 0$. This is valid for $x_1$, $x_2$, …, $x_n$ being a sample of $n$ independent and identically distributed realizations of a random variable $X$ following some common distribution function $F$ (Coles, 2001).

A number of approaches have been proposed for selecting an appropriate threshold. Coles (2001) suggests to let the selection be guided by graphical diagnostics about bias (mean excess) and stability of the scale and shape parameter. Despite these criteria are well justified from a theoretical point of view, its application involves substantial elements of subjectivity leading to ambiguous results (Scarrott and MacDonald, 2012; Northrop and Coleman, 2014). To overcome this problem, we employed the deterministic square root rule $k = \sqrt{n}$ (Ferreira et al., 2003) for pre-selecting the threshold level in an objective way, using the $k^{th}$ upper order statistic as a threshold, which is related to the total time series length $n$. Albeit this rule does not properly account for threshold uncertainty on subsequent inferences (Scarrot and MacDonald, 2012), it satisfies the intermediate sequence of order statistics that formally ensures tail convergence (Leadbetter et al., 1983). The so-obtained threshold was subsequently validated by the graphical criteria of Coles (2001) for bias and parameter stability.



## 2.3 Dealing with non-stationarity and dependency

Extreme value theory assumes that data are independent and identically distributed (Coles, 2001; Gilleland and Katz, 2011; Katz, 2010; Katz, 2013; Cheng et al., 2014). To test for non-stationarity in the expected value we perform separate Mann-Kendall trend tests (Mann, 1945; Kendall, 1976; Gilbert, 1987) at a significance level of $\alpha = 0.05$ (Zhang et al., 2004) for

the extreme value series of each meteorological indicator. In case of significant trends, detrending was performed with respect to the last year of the time series (i.e. 2015). The trend-corrected estimation of a meteorological indicator $z$ at time $t$ is obtained as

$$\hat{z}_t = y_t - \hat{y}_t + \hat{y}_{2015} \tag{3}$$

where $y_t$ is the measurement at time $t$ and $\hat{y}_t$ is the trend at time $t$ obtained from the linear trend model

$$\hat{y}_t = \beta_0 + \beta_1 t \tag{4}$$

with intercept $\beta_0$ and slope $\beta_1$, and $\hat{y}_{2015}$ being the trend estimate for 2015.

For climate variables independence of data is usually a minor issue for the annual maxima approach as multi-annual dependencies are usually low for most climates (Madsen et al., 1997a; Katz et al., 2002). Regarding the threshold excess method, threshold exceedances on consecutive days will likely violate the assumption of independence. Dependent values in the threshold excess series are eliminated by a declustering procedure that consists in removing threshold exceedances within the autocorrelation length on both sides of the local maxima (Jarušková and Hanek, 2006). Based on sensitivity analysis an

autocorrelation window of 5 days was chosen for the three temperature indicators, while a window of 3 days was chosen for the accumulated daily precipitation.

## 2.4 Parameter estimation

Once the extreme value series is available, a theoretical distribution needs to be fitted. Two different methods of parameter estimation are used within the scope of the present analysis.

The first method, maximum-likelihood estimation (MLE), was formally introduced by Fisher in the early 20[th] century (Fisher, 1912; Aldrich, 1999; Hald, 1999). Let $x_1$, $x_2$, …, $x_n$ be a sample of $n$ independent and identically distributed realizations of a random variable with the unknown probability density function $f(x|\theta_0)$. As the true value of the parameter vector $\theta_0$ is unknown, an estimate $\hat{\theta}$ which is as close to $\theta_0$ as possible is found by maximizing the likelihood function

$$L(\theta) = \prod_{i=1}^{n} f(x_i|\theta) \tag{5}$$

i.e. by maximizing the accordance of the extreme value model with the observed data (Coles, 2001).

The second method, L-moments estimation (LMOM), evolved from modifications of probability weighted moments of Greenwood et al. (1979). They are linear combinations of first order statistics and are hence more robust to measurement errors or sampling uncertainty than conventional moments (Hosking, 1990). The $r^{th}$ population L-moment of a random variable $X$ is defined as


$$\lambda_r \equiv r^{-1} \sum_{k=0}^{r-1} (-1)^k \binom{r-1}{k} EX_{r-k:r} \,, \qquad r = 1, 2, \dots \tag{6}$$

As compared to MLE, L-moments are superior for fitting GEV distributions in terms of bias and variance, in particular for small sample sizes (Hosking et al., 1985).

As far as reliability of the fitting results is concerned, confidence intervals play a major role for assessing uncertainty. The most common way to derive a $(1 - \alpha)$ confidence interval for a particular component $\theta_i$ of a parameter vector $\theta$ is by using the formula $\hat{\theta}_i \pm z_{\alpha/2} \times \sigma/\sqrt{n}$, with $\hat{\theta}_i$ denoting the estimate for $\theta_i$, $z_{\alpha/2}$ indicating the $\alpha/2$ quantile of the standard normal distribution and $\sigma/\sqrt{n}$ indicating the standard error of the estimate.

The approach assumes Gaussian distributed parameter estimators, which may be inappropriate for extreme value distributions. For LMOM estimators resampling methods have been recommended (Burn, 2003). Thus, nonparametric bootstrapping with 500 iterations was applied in this study. MLE offers a more accurate method for deriving confidence intervals based on the profile likelihood (Coles, 2001). The profile log-likelihood for $\theta_i$ is defined as

$$L_p(\theta_i) = \max_\delta L(\theta_i, \delta) \tag{7}$$

where $\delta$ denotes all components of parameter vector $\theta$ excluding $\theta_i$. That is, for each value of $\theta_i$, $L_p(\theta_i)$ is the maximized log-likelihood over all remaining elements of $\theta$.

## 2.5 Assessment method

There are various performance measures that are regularly employed in model evaluation, including the root mean squared error (RMSE) and the mean absolute error (MAE). These metrics provide a comprehensible and objective basis regarding the assessment of the fitted functions.

In addition, most events of the extreme value series are only moderate and these will have an overly excessive influence on the performance measure. In order to specifically assess the accuracy of the fitted models for higher quantiles (i.e. for larger return periods), we propose conditional variants of the root-mean-square deviation (CRMSE$_T$) and mean absolute error (CMAE$_T$). These metrics are conditional on the return period $T$ of the underlying data and specifically consider the upper tail of the fitted functions. Using Weibull plotting positions as empirical probability estimator (Weibull, 1939; Makkonen, 2005), these measures are defined as

$$\text{CRMSE}_T = \sqrt{\frac{\sum_{i=1}^{n} (\hat{y}_i - y_i)^2}{n_t}} \, \forall y_i : \left[ -\frac{1}{\ln\left(\frac{m}{N+1}\right)} \right] \geq T \tag{8}$$

$$\text{CMAE}_T = \frac{\sum_{i=1}^{n} |\hat{y}_i - y_i|}{n_t} \, \forall y_i : \left[ -\frac{1}{\ln\left(\frac{m}{N+1}\right)} \right] \geq T \tag{9}$$

where $\hat{y}_i$ denotes the model prediction or the $i^{th}$ element of the extreme value series, $y_i$ is its observed value, $m$ is its order statistic (with $m = 1$ for the minimum and $m = N$ for the maximum), and $n_T$ is the number of elements with an empirical





return period greater than $T$. Hence, the conditional performance measures are calculated by using only the residuals of observations and theoretical distribution above some relevant return level $T$. In this study, $T = 10$ years has been chosen and the $CRMSE_{10}$ and $CMAE_{10}$ are calculated.

In addition to the goodness-of-fit analysis we performed graphical diagnosis of the extreme value series and the fitted distributions in quantile plots. For AMS, plotting of empirical distributions is straightforward. The return level (i.e. magnitude) $z_T$ of each observed extreme event is plotted against its return period (i.e. recurrence interval) $T = 1/(1 - P)$, using Weibull plotting positions as an estimator of empirical recurrence probability $P$. For AMS, the $T$-year return level is obtained using the quantile function of the GEV:

$$z_{T,AMS} = \begin{cases} \mu - \dfrac{\sigma}{\xi}\big[1 - \{-\ln(P)\}^{-\xi}\big] & \text{for } \xi \neq 0 \\ \mu - \sigma \ln\{-\ln(P)\} & \text{for } \xi = 0 \end{cases} \tag{10}$$

with parameters according to Eq. 1. In the case of PDS the return period $T$ needs to be transformed from an observation scale to an annual scale by taking into account the number of threshold exceedances within the observation period (Coles, 2001). Hence, the $T$-year return level is obtained from the quantile function of the GP by:

$$z_{T,PDS} = u + \dfrac{\sigma}{\xi}\big[(Tn\zeta_u)^{\xi} - 1\big] \tag{11}$$

where $u$ is the threshold, $n$ is the number of observations per year, $\zeta_u$ is the sample proportion of threshold exceedances, and remaining parameters according to Eq. 2. The so obtained return levels were used for synoptic plotting of AMS and PDS.

## 3 Results

### 3.1 Non-stationarity

Linear trends were considered by incorporating dependency on time by means of precedent detrending within model estimation. Most of the temperature hot spots showed a significant change over time in at least one of temperature indicators. The observed positive temperature trends lead to both an increase of daily maximum temperatures and at the same time to an increase of daily minimum temperatures. As illustrated by Fig. 2, the consequence of incorporating a trend model in the analysis are non-stationary return levels that refer to a specific time. We will give results for the end of the observation period.





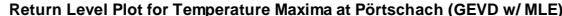

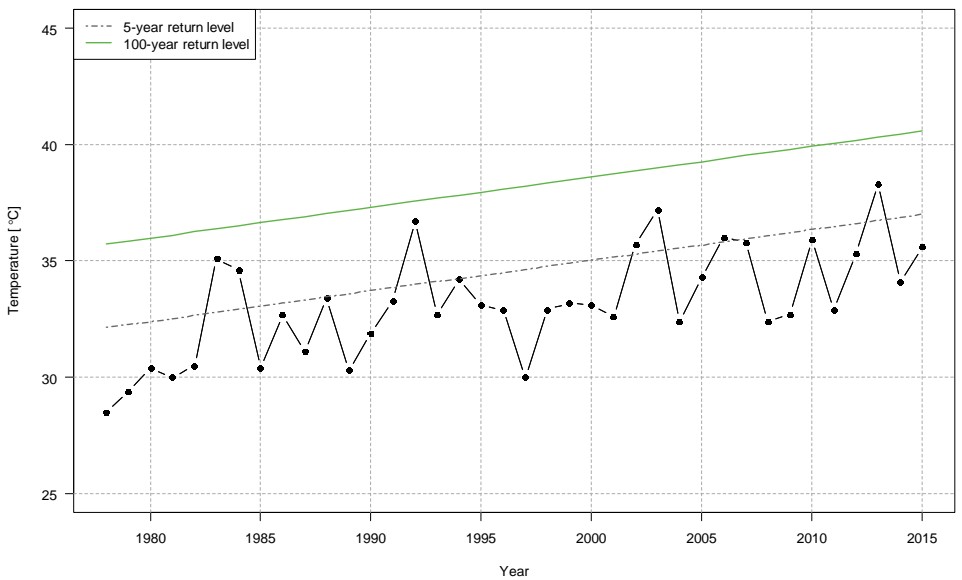

**Figure 2: Return level plot of temperature maxima at Pörtschach (Carinthia) with linear trend correction. The trend is visible in the lines depicting the 5-year return level (gray dashed line) and the 100-year return level (green solid line). This is an illustrative example of temperature trends that are commonly observed at the selected stations for both temperature maxima (increasing trend) and temperature minima (decreasing trend).**

For precipitation, non-stationarity seems less important than for temperature indicators: About 85 % of the hot spots of our study area showed no trend in the annual extremes. This is consistent with the expectation of the Austrian Panel of Climate Change (APCC, 2014) that climate impacts on precipitation will mainly lead to seasonal shifts rather to changes in total annual precipitation.

## 3.2 Parameter estimation method

The two approaches have been tested for the four meteorological indicators. In summary, it becomes apparent that the relative performances of MLE and LMOM are strongly situation-dependent. For instance, while the return level plots for temperature maxima at *Schwechat* in the eastern lowlands show that the function fitted on the basis of LMOM behaves more robust, which appears to be beneficial in this case (Fig. 3), return level plots of daily rainfall at *Brenner* on the Austrian-Italian border indicate that the less robust MLE offers better fit for higher quantiles (Fig. 4).




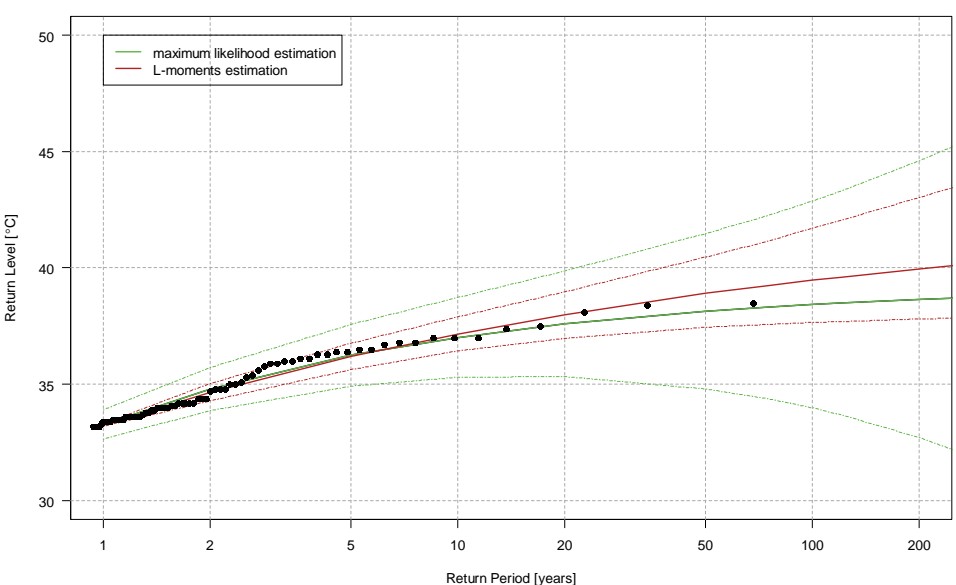

**Figure 3: Return level plot of temperature maxima at *Schwechat*. Return level estimation is based on the threshold excess approach with two different parameter estimation methods (MLE and LMOM-estimation). Solid lines show the mean estimate, while dashed lines indicate the 95% confidence intervals for the fitted functions.**

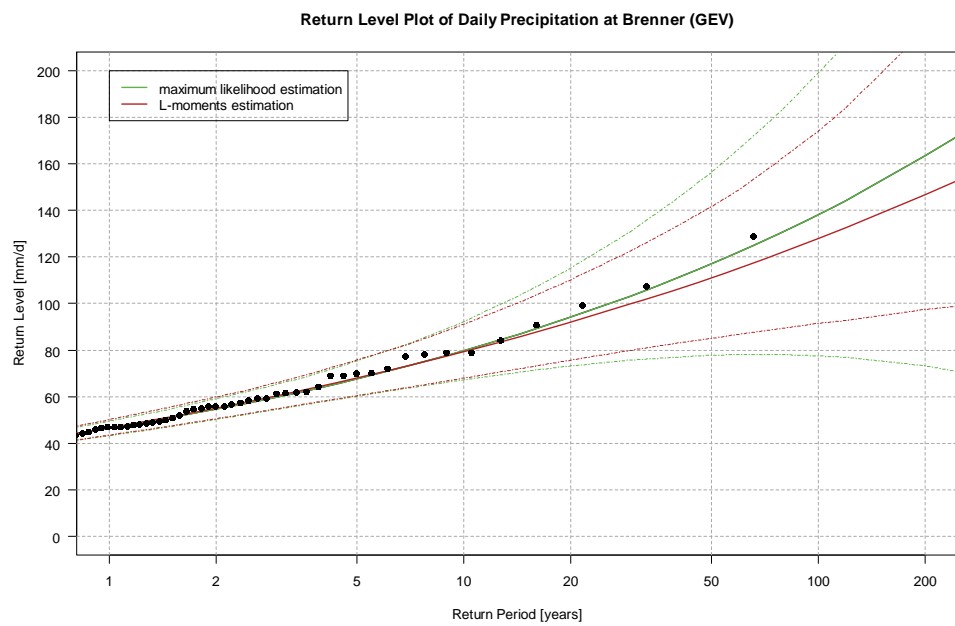

**Figure 4: Return level plot of daily rainfall events at Brenner. Return level estimation is based on the block maxima approach with two different parameter estimation methods (MLE and LMOM-estimation). Solid lines show the mean estimate, while dashed lines indicate the 95% confidence intervals for the fitted functions.**





Tab. 1 summarizes the overall goodness-of-fit for the 100 climate records (25 stations x 4 indicators) assessed in this study for the AMS approach. LMOM performed better in 69 % of the cases when assessed by the RMSE, and in 94 % when assessed by the MAE (note that for 100 climate records one percent corresponds to one record). Since the MAE favors overall model accuracy and gives little weight to outliers with large errors, the better overall fit achieved by LMOM nicely illustrates the greater robustness of this method. These differences apply to most individual meteorological indicators. The sole exception is daily minimum temperature, which yields similar success rates of MLE and LMOM for both goodness-of-fit measures. This is attributable to several larger residuals in these time series.

**Table 1: Comparison of parameter estimation methods for the AMS approach based on goodness-of-fit measures RMSE and MAE. Numbers indicate success rates (% of records) of MLE and LMOM.**

| Indicator | RMSE (MLE) | RMSE (LMOM) | MAE (MLE) | MAE (LMOM) |
|---|---|---|---|---|
| Precipitation | 7 | 18 | 4 | 21 |
| $T_{min}$ | 13 | 12 | 1 | 24 |
| $T_{max}$ | 5 | 20 | 0 | 25 |
| $T_\Delta$ | 6 | 19 | 1 | 24 |
| **Total** | **31** | **69** | **6** | **94** |

The relative performances turned out to be more balanced with respect to the PDS approach. As indicated by Tab. 2, MLE performed better in 56 % and 53 % of the cases when judged by the RMSE and MAE, respectively. Again, daily minimum temperature deviates from the general picture, by showing clear advantages in favor of LMOM-estimation in this case.

**Table 2: Comparison of parameter estimation methods for the PDS approach based on goodness-of-fit measures RMSE and MAE. Numbers indicate success rates (% of records) of MLE and LMOM.**

| Indicator | RMSE (MLE) | RMSE (LMOM) | MAE (MLE) | MAE (LMOM) |
|---|---|---|---|---|
| Precipitation | 14 | 11 | 13 | 12 |
| $T_{min}$ | 9 | 16 | 12 | 13 |
| $T_{max}$ | 17 | 8 | 14 | 11 |
| $T_\Delta$ | 16 | 9 | 14 | 11 |
| **Total** | **56** | **44** | **53** | **47** |

Apart from the overall goodness-of-fit it is interesting to assess how the fit depends on the return period of events. This has been done by visual inspection of the distribution plots, such as the examples shown in Fig. 3 and Fig. 4. In most cases there were only minor differences between MLE and LMOM when considering return levels below 10 years, but often considerable differences for larger return periods. For the 100-year events, e.g., results of the temperature indicators differed by about 0.5 °C on average, and by up to 2 °C for single stations. With maximum differences around 10 mm/d, the 100-year precipitation events showed even greater variation.



As the objective of extreme value analysis is usually related to return periods of 10 years or more, we specifically assessed the performance of the extreme upper tail of the distribution by the conditional goodness-of-fit measures $CRMSE_{10}$ and $CMAE_{10.}$ Results indicate again a favorable performance of LMOM-method for AMS series (Tab. 3), when judged by the $CRMSE_{10}$ (58 %) and the $CMAE_{10}$ (62 %).

**Table 3: Comparison of parameter estimation methods for the AMS approach based on conditional goodness-of-fit measures $CRMSE_{10}$ and $CMAE_{10}$. Numbers indicate success rates (% of records) of MLE and LMOM.**

| Indicator | $CRMSE_{10}$ (MLE) | $CRMSE_{10}$ (LMOM) | $CMAE_{10}$ (MLE) | $CMAE_{10}$ (LMOM) |
|---|---|---|---|---|
| Precipitation | 11 | 14 | 11 | 14 |
| $T_{min}$ | 11 | 14 | 10 | 15 |
| $T_{max}$ | 12 | 13 | 8 | 17 |
| $T_{\Delta}$ | 8 | 17 | 9 | 16 |
| **Total** | **42** | **58** | **38** | **62** |

In contrast, results for the PDS showed, again, a slight advantage of MLE when assessed with the goodness-of-fit measures for the conditional variants. Both measures indicate a preference towards MLE in 58 % of the cases. The better performance

of the MLE method is against the expectation based on robustness and will be examined in more detail in the following section.

**Table 4: Comparison of parameter estimation methods for the PDS approach based on conditional goodness-of-fit measures $CRMSE_{10}$ and $CMAE_{10}$. Numbers indicate success rates (% of records) of MLE and LMOM.**

| Indicator | $CRMSE_{10}$ (MLE) | $CRMSE_{10}$ (LMOM) | $CMAE_{10}$ (MLE) | $CMAE_{10}$ (LMOM) |
|---|---|---|---|---|
| Precipitation | 14 | 11 | 14 | 11 |
| $T_{min}$ | 9 | 16 | 10 | 15 |
| $T_{max}$ | 19 | 6 | 19 | 6 |
| $T_{\Delta}$ | 16 | 9 | 15 | 10 |
| **Total** | **58** | **42** | **58** | **42** |

**3.3 Extreme value selection**

Tab. 5 presents the relative performances of AMS and PDS approaches based on the two parameter estimation methods. Albeit overall results show advantages for the AMS approach in terms of goodness-of-fit for the upper tail of the underlying distributions, results largely depend on the underlying meteorological indicators. While precipitation and daily maximum temperature difference offer a better fit when using GEV distributions of AMS, GP distributions of PDS appear better suited for modelling daily temperature maxima and minima.





**Table 5: Comparison of AMS and PDS approach based on conditional goodness-of-fit measures CRMSE10 and CMAE10 for two parameter estimation methods MLE and LMOM. Numbers indicate success rates (% of records) of approaches.**

| Indicator | Fitting Method | $CRMSE_{10}$ (GEV) | $CRMSE_{10}$ (GP) | $CMAE_{10}$ (GEV) | $CMAE_{10}$ (GP) |
|---|---|---|---|---|---|
| Precipitation | MLE | 18 | 7 | 19 | 6 |
| Precipitation | LMOM | 19 | 6 | 20 | 5 |
| $T_{min}$ | MLE | 9 | 16 | 8 | 17 |
| $T_{min}$ | LMOM | 10 | 15 | 10 | 15 |
| $T_{max}$ | MLE | 10 | 15 | 11 | 14 |
| $T_{max}$ | LMOM | 13 | 12 | 14 | 11 |
| $T_{\Delta}$ | MLE | 16 | 9 | 17 | 8 |
| $T_{\Delta}$ | LMOM | 17 | 8 | 16 | 9 |
| **Total** | | **112** | **88** | **115** | **85** |

5    To perform a direct comparison, Fig. 5 presents the deviations between return levels derived via AMS and PDS approach for the four meteorological indicators. A common pattern regarding the magnitude of the estimated return levels can be observed. While GP estimates seem to result in higher return levels for lower return periods (indicated by negative deviations), this behavior changes to the opposite for higher return periods. This issue will be further explored in the discussion section. Nonetheless, it shall be noted that daily minimum and maximum temperature do not fully fit into these

10   patterns. Albeit maximum temperature shows the same tendencies as precipitation, the PDS always yields higher return levels than the AMS, suggesting that differences mainly occur at higher return periods. Temperature minima, however, show a rather constant overestimation (i.e., underestimation of negative magnitude) of PDS compared to AMS regardless of the frequency of events.





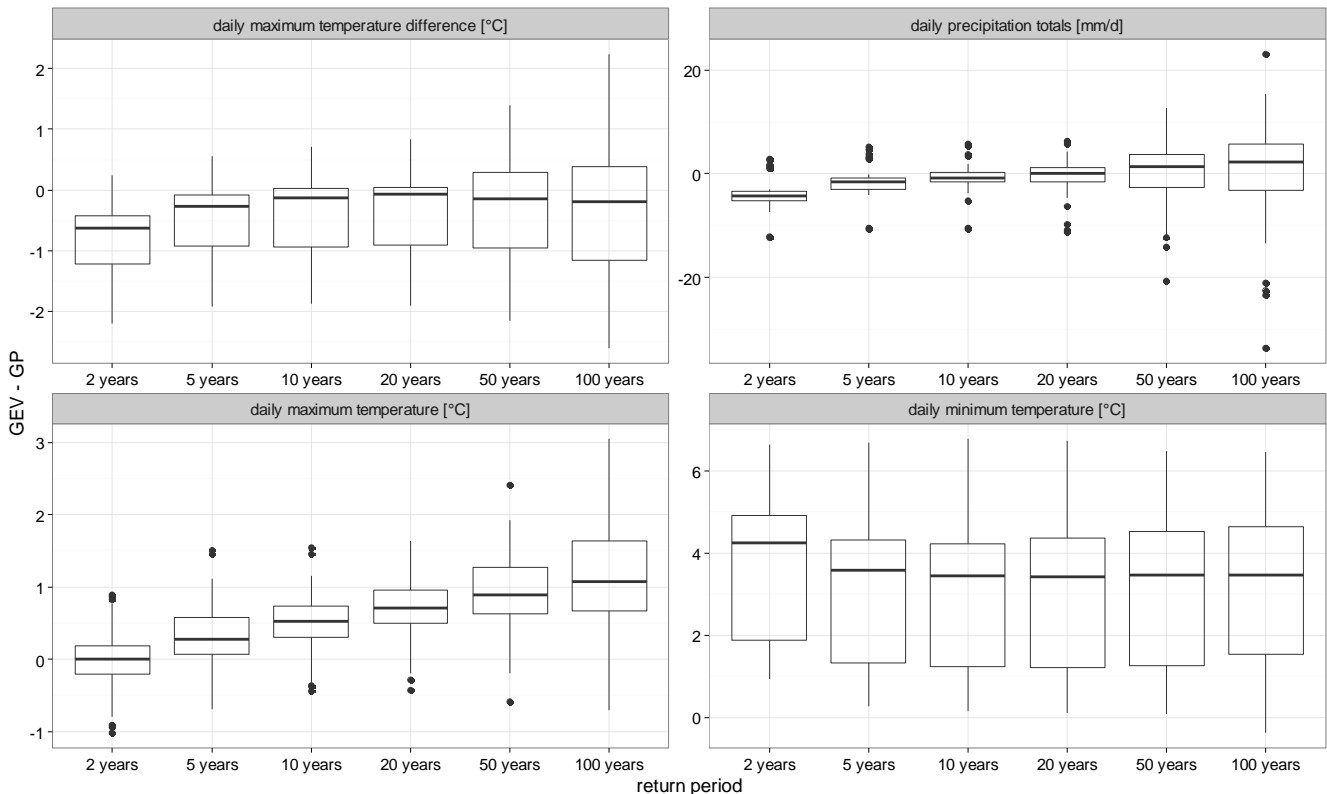

**Figure 5: Differences in estimated return levels between GEV and GP models for six selected return periods. These differences are calculated by subtracting the GP estimate from the GEV estimate, given the same parameter estimation method. This results in $n = 50$ observations per boxplot.**

5   Finally, Tab. 6 summarizes the success rates of all methods based on $CRMSE_{10}$. Results show an overall advantage of using L-moments estimation as compared to MLE. As far as the two different methods of extreme value selection are concerned, the AMS approach seems to slightly outperform the threshold excess approach in this study. While results are basically quite balanced between all four methods, AMS fitted on the basis of LMOM estimation turned out to yield the best results in about 35% of all cases.





**Table 6: Success rates of methods according to CRMSE$_{10}$. The bold value in the center of each field indicates the overall count. The four smaller numbers in the corners display the counts with respect to temperature minima (top left), temperature maxima (top right), temperature difference (bottom left) and precipitation totals (bottom right). Bold values indicate better performance.**

| | | Distribution | | |
|---|---|---|---|---|
| | | GEV | GP | Total |
| **Fitting method** | **MLE** | 3  4 <br> **19** <br> 4  8 | 5  **12** <br> **19** <br> 1  1 | **38** |
| | **LMOM** | 7  5 <br> **35** <br> **12**  **12** | **10**  4 <br> **27** <br> 8  4 | **62** |
| | **Total** | 54 | 46 | 100 |

## 4 Discussion

We compared the relative merits of the block maxima method and the threshold excess approach. In addition, two different fitting methods have been contrasted. This results in four possible combinations of extreme value model parameter estimation, all of which have certain strengths and weaknesses. Concerning the fit of the distributions to sample, we found a slight advantage of using LMOM instead of MLE, especially in combination with AMS. For PDS there was a slight advantage of using MLE. But overall, the differences were not huge.

The conditional assessment of the individual deviation between return levels of AMS and PDS yielded deeper insight in the relative performances of methods. Most importantly, we found systematic deviations between both approaches (Fig. 5): For low return periods (non-extreme events) the PDS approach tends to overestimate return levels as compared to the AMS approach. An opposite behavior was found for high return levels (extreme events). To assess the reasons for this systematic behavior, we selected four example series that represent extreme cases, where results of approaches differ significantly.

The first two examples are daily precipitation at Sankt Michael (Fig. 6a) and Brenner (Fig. 6b), where extreme value series deviate from the ideal, smooth behavior of a homogeneous extreme value series. These fluctuations point to either measurement errors or process heterogeneity that will introduce uncertainty into extreme value analysis. In the case of Sankt Michael, the most extreme events appear as outliers that deviate from the general behavior of the sample. In general, LMOM will give lower weight to such leverage points but this seems not the case here where the GP fitted by LMOM seems more

attracted. A plausible explication would be that the upper-tail behavior is resulting from the attraction of the distribution at the lower end, because of the limited flexibility of the GP. In the case of Brenner, the extreme values seem to follow the same distribution than the remaining sample so one would have more confidence in the validity of these values. However, extreme values are always prone to higher uncertainty than the remaining sample. The MLE estimate gives more weight to these values and shows a better fit at the upper tail in this case, whereas LMOM gives less weight to these values, and makes





visible that they are not perfectly following the shape of the entire distribution. The choice of the parameter estimation method will finally depend on the weight one tends to give to the extreme values as compared to the remaining sample.

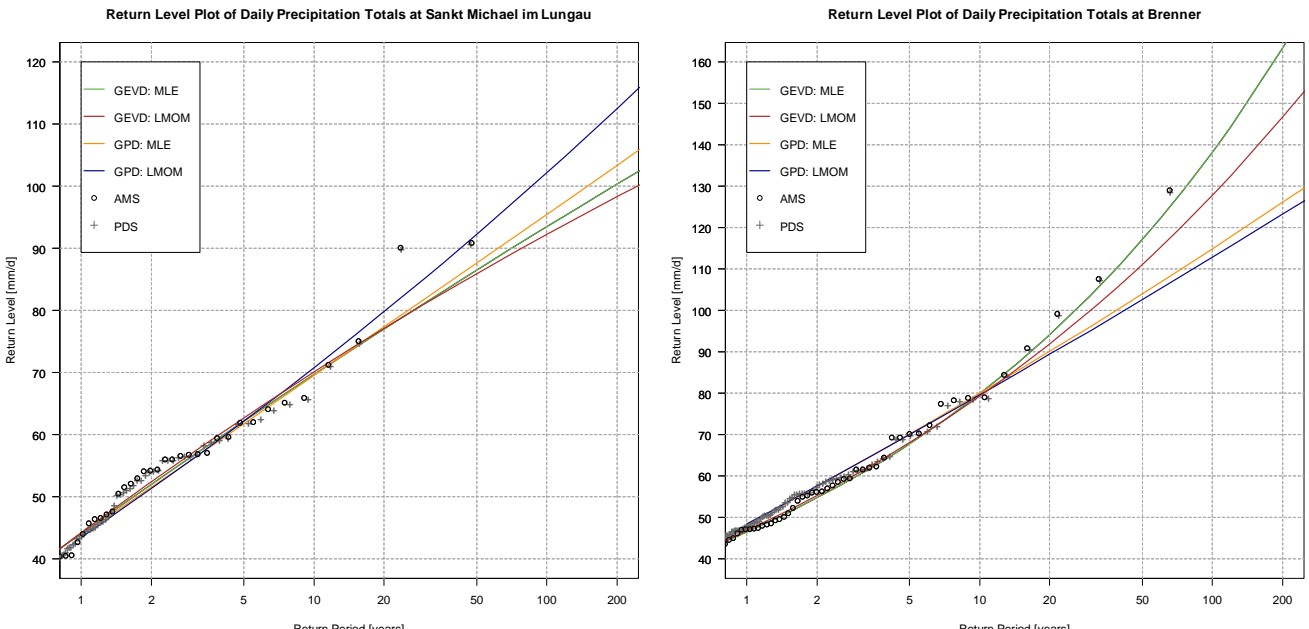

**Figure 6: Return level plots of daily rainfall events at the hot spot in (a)** *Sankt Michael im Lungau***, which is located in the Central Eastern Alps and (b)** *Salzburg***, located at the northern edge of the Alps. Return level estimation is based on the block maxima approach and on the threshold excess approach with two different parameter estimation methods (MLE and LMOM-estimation). Based on the $CRMSE_{10}$, GP fitted on the basis of LMOM-estimation was found to be the most appropriate method for Sankt Michael, while GEV with MLE was found to be most suitable in Salzburg. Please note that functions are plotted without associated confidence intervals for the sake of clarity.**

It is also interesting to analyze extreme cases where AMS and PDS methods yield contrasting results (Fig. 7). When focusing on the empirical distributions, we observe that only the highest events (three in the case of *Bruck an der Mur*, and two in the case of *Graz*) have almost identical empirical probabilities in both extreme value series. At the lower end, we observe that there are several events in the AMS below the threshold level of PDS, which fit well to the distribution of the higher values so we find no evidence to exclude them from the analysis. The shift in the distribution can therefore be regarded as an effect of threshold level selection, which determines the lower end and therefore the shape of the lower part of the PDS distribution. Between the undisturbed upper part and the disturbed lower part a breakpoint at $T = 15$ years in the PDS is clearly visible from the robustly fitted GPD using the LMOM method. This illustrates an inherent danger of the PDS approach: An inappropriate threshold may entail considerable biases that outperform the possible gain of information by the method by far. This was neither visible from the square-root criterion nor from the graphical diagnosis (residual life plot, Fig. 8) which yielded indeed almost no bias in both cases (in the case of *Bruck an der Mur*, mean excess = 2.99 for the threshold of -17.1 °C, and in the case of *Graz*, mean excess = 1.82 for threshold of 32.1 °C).





Similar shifts may arise if the extreme value series contains dependent events. Non-extreme events are generally more likely to cluster than extreme events because they are generated by exceptional process combinations which are unlikely to occur more often during one extreme weather situation. Thus, dependencies may possibly affect all parts (but more likely the lower part) of the distribution apart from the maximum, which remains unchanged. In consequence, the empirical distribution is

stretched at the lower tail (shifted to the left), with similar consequences on lower and upper tail as described for the case of data uncertainty and leverage points. Such artifacts are difficult to detect in quantile plots of one extreme value series alone, but are often visible from direct comparison of AMS and PDS approaches. Albeit both AMS and PDS may be affected by dependency of events, AMS behaves more robust since it selects only one event per year.

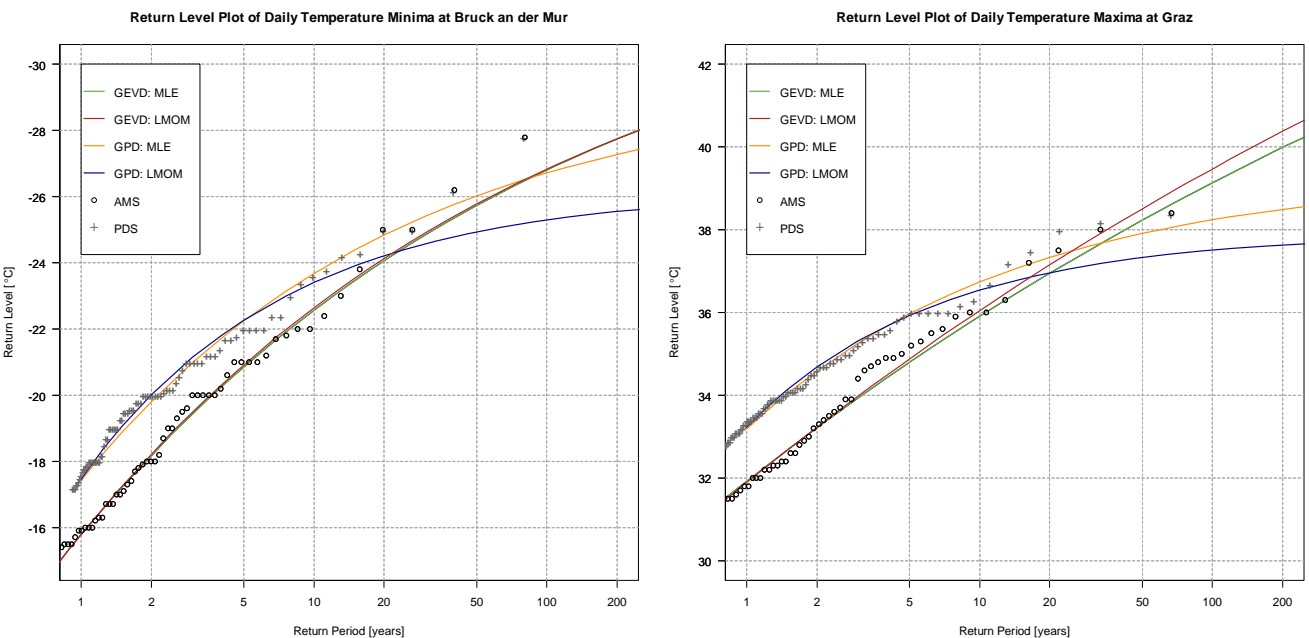

**Figure 7: Return level plots of (a) temperature minima at *Bruck an der Mur*, and (b) temperature maxima at *Graz*. Return level estimation is based on the block maxima approach and on the threshold excess approach with two different parameter estimation methods (MLE and LMOM-estimation).**

We did not expect these findings, which contradict to the spirit of most existing studies that aimed to recommend the best performing method for a variable or situation. We recommend performing both approaches, as their synoptic assessment by

means of diagnostic plots together with overall and conditional goodness-of-fit measures offers a more complete diagnosis of the quality of extreme series and the resulting distributions.

Concerning the parameter estimation method, there are also benefits and disadvantages that have to be balance against each other. MLE has some merit with respect to calculating reliable confidence intervals via profile likelihood. Confidence intervals for estimation via LMOM were derived with non-parametric bootstrapping, which is arguably less trustworthy for

indicating the uncertainty of the estimates. However, LMOM-estimation has been shown to yield more robust estimation results for small sample sizes (Hosking et al., 1985; Hosking and Wallis, 1987), which can be especially beneficial when





analyzing environmental data like temperature or precipitation indicators, which are derived from raw measurements at meteorological measuring stations. Regarding the overall results, LMOM-estimation turned out to offer a better fit than MLE, which is consistent with previous findings (Hosking et al., 1985; Hosking and Wallis, 1987; Bezak et al., 2014).

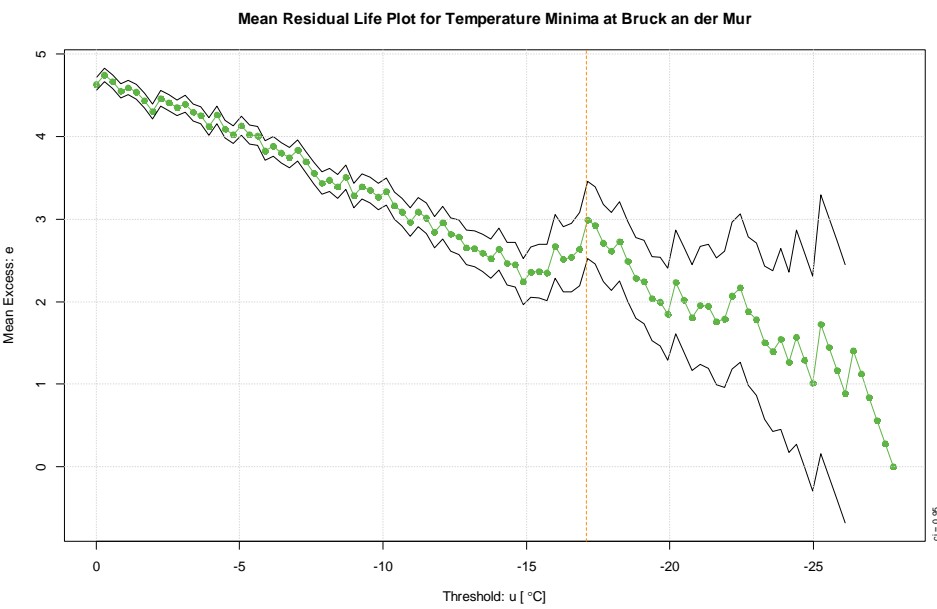

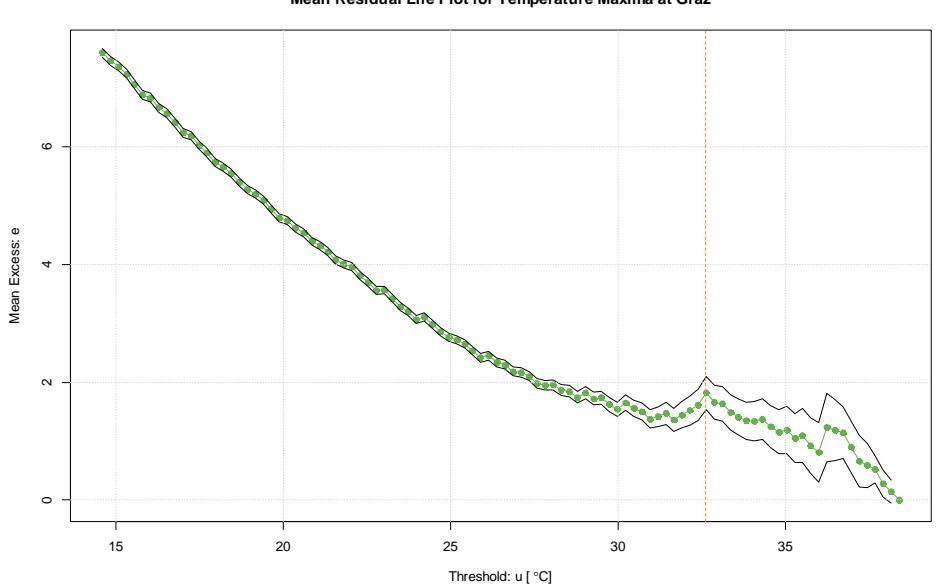

**Figure 8: Mean residual life plots of (a) temperature minima the hot spot in *Bruck an der Mur* and (b) temperature maxima at *Graz*. Black lines indicate the 95% confidence interval for the mean excess and orange lines indicate the threshold selected by means of the square root rule.**



Concerning the comparison based on the goodness-of-fit of the distributions it shall be noted that a formal comparison of the two extreme value selection approaches is not straightforward. Measures of goodness-of-fit are not fully conclusive, as the underlying extreme value series are derived by different methods and thus are not directly comparable. Our analysis demonstrates that the choice between these approaches has to be based on the statistical properties of the extreme value

series, which are related to the indicators under consideration and on data availability. The conditional measures proposed in this paper help to perform a more specific assessment for extreme events, but they are also not a remedy to overcome this problem. They are a way to assess the goodness of fit at the upper tail of the distribution and facilitate the comparison between AMS and PDS. These metrics can assist, but not substitute careful analysis of assumptions. We show that contrastive plotting methods can strongly support these analyses.

While the methodology of this study can be easily generalized and extended to cover other environmental variables, two possible limitations have to be discussed. Firstly, seasonality of temperature and precipitation extremes has not been taken into account. While maximum/minimum temperatures will always occur in the same season, which will factor out any seasonal heterogeneity, this is not genuinely the case for extreme precipitation events, where different seasonality of occurrence may be linked with different processes (Hundecha et al., 2009). In order to account for seasonal effects, a

common approach is to split the events into process-homogeneous subsets based on seasonality (e.g. Laaha and Blöschl (2006) for low streamflows), on a typology of processes (e.g. Merz and Blöschl (2003) for floods based on rainfall types and catchment preconditions) or on a temporal stratification (e.g. Méndez et al. (2008) for wave height and Maraun et al. (2009) for heavy precipitation). For each subset extreme value analysis is performed separately, leading to process-specific return levels, such as summer and winter low flows in the case of minimum discharges. These quantities may be combined by a

mixed distribution model to yield overall return levels (e.g. Hundecha et al., 2009). For further discussion of modelling dependent and non-stationary time series extremes it is referred to Chavez-Demoulin and Davison (2012).

Secondly, threshold selection in the threshold excess method is a legitimate subject for debate. In recent years, efforts have been made to overcome the problem of visual threshold selection, e.g. by robust threshold selection (Dupuis, 1999), additional likelihood-based procedures for supplementing visual diagnostics (Wadsworth and Tawn, 2012; Wadsworth,

2016), Bayesian approaches (Tancredi et al., 2006; Lee et al., 2014) and extreme value mixture models (MacDonald et al., 2011). In addition, attempts were made to develop more automated approaches for extreme value threshold estimation, including the automated threshold selection approach by Thompson et al. (2009), the multiple threshold method by Deidda (2010) and the automatic threshold and run parameter selection by Fukutome et al. (2015). Several automated threshold selection methods (ATSM by Thompson et al., 2009; MTM by Deidda, 2010) which have been tested for the time series

under consideration yielded dissatisfying and inconsistent thresholds: Threshold values for time series which exhibit similar empirical distributions varied considerably and it was noticed that results were depending strongly on the range over which the functions are fit as well as the number of breaks set within this range. While certain patterns of convergence were found using brute force methods, it is argued that these procedures somehow replace the threshold selection problem with that of



selecting an appropriate range and an appropriate number of breaks. Therefore, the application of the square root rule in combination with graphical diagnostics is a feasible approach that led satisfactory results in the present study.

## 5 Conclusion

We compared statistical methods for extreme value analyses based on four climate indicators related to daily precipitation and temperature. While the indicators were selected for studying the exposure of road infrastructure to extreme weather events, the assessments are equally relevant for a range of other environmental variables including meteorological and hydrological quantities. We first analyzed the goodness-of-fit of distributions to extreme value series consisting of annual maxima (AMS) and threshold exceedances (PDS) using two parameter estimation methods.

Results for the parameter estimation methods vary considerably between stations and approaches. For the AMS approach, LMOM yielded, on average, better fitted distributions than MLE. The goodness-of-fit turned out to be more balanced with respect to the PDS approach, with a slight advantage of MLE. In most cases there were only minor differences between MLE and LMOM when considering return levels below 10 years, but often considerable differences for larger return periods.

Concerning extreme value selection, the relative performance of AMS and PDS approaches vary between meteorological indicators. For precipitation and temperature difference the AMS data outperformed the PDS approach. For temperature maxima and minima the PDS approach appeared better suited.

Regarding goodness-of-fit for extreme events that are typically used as design-values ($T$ of 10 years and more), results show an overall advantage of using L-moments estimation as compared to MLE, and that the AMS approach slightly outperforms the threshold excess approach. The AMS fitted on the basis of LMOM estimation method performed better than all other combinations of approaches in this study.

We further examined the conditional performances of AMS and PDS approaches with respect to the return period in more detail. From conditional performance measures and synoptic plots, we found systematic deviations between AMS and PDS approaches. For low return periods (non-extreme events) the PDS approach tends to overestimate return levels as compared to the AMS approach, whereas an opposite behavior was found for high return levels (extreme events). The assessment of extreme cases where approaches differed significantly suggests that this behavior may be related to two factors, sampling uncertainty and threshold selection.

Regarding sampling uncertainty, we found that outliers may not only attract the distribution at the tail where they occur, but they may also bend the curve at the opposite tail as a consequence of limited flexibility of the extreme value distributions. Such leverage effects can be handled by careful inspection of quantile plots. Regarding threshold selection, the analysis of extreme cases within the data set revealed that an inappropriate threshold may lead to considerable biases that may outperform the possible gain of information from including additional extreme events by far. Selecting a high threshold will determine the lower end of the extreme value distribution whereas the upper tail remains unchanged. This may introduce an inflection point in the distribution, which is against its ideal shape according to extreme value theory, resulting in poor





estimates of the theoretical distribution. This effect was neither visible from the square-root criterion, nor from the graphical diagnosis (mean residual life plot) which yielded indeed no atypical biases for the analyzed cases. Similar effects may arise when the extreme value series contains dependent events that may stretch the empirical distribution at the part where they occur. These findings where against our expectations that the estimation of the theoretical distribution will greatly profit

from the gain of information that is provided by the PDS approach.

We emphasize the reliable extreme value statistics require controlling for sample effects in order to avoid biased models. In our study, the differences and relative merits of methods were best visible from a direct comparison of AMS and PDS approaches. We therefore recommend performing both analyses and carefully analyze the fit of distribution relative to the respective sample and relative to each other, by means of synoptic quantile plots. This will make the analyses more robust, in

cases where threshold selection and dependency introduces biases to the PDS approach, but also in cases where the AMS contains non-extreme events that may introduce similar biases. For assessing the performance of extreme events we recommend conditional performance measures such as $CRMSE_{10}$ and $CMAE_{10}$ in addition to unconditional indicators.

### Acknowledgements

The paper is a contribution to UNESCO's FRIEND-Water program. The authors would like to thank the Austrian Climate

Research Program ACRP for financial support through the project DALF-Pro (GZ B464822). We thank the Central Institution for Meteorology and Geodynamics (ZAMG) for providing meteorological data.

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
