# Peer review of "Extreme weather exposure identification for road networks – a comparative assessment of statistical methods"

_Natural Hazards and Earth System Sciences, 2016_

## Referee Comment (RC1) · Anonymous Referee #1 · 26 Dec 2016

General comments: A very interesting work on the theoretical and practical aspects of estimating the retirn period of extreme meteorological events, useful in a very wide range of applications. The theme of manuscript is well within the scope of NHESS since it sums up and compares the major methodologies currently used for extreme values modelling with a specail focus on extreme weather risk assessment. The title and the abstract are informative and inviting to the prospective reader. The presentation of the methods is comprehensive, easily understandable by a wider audience, and their application rigorous , adhering to best statical practice guidelines. Appropriate references are provided wherever needed. Figures and tables are well placed and commented, without any redundancy. The results reached by the authors are clearly

and methodicaly presented and readily expoitable by the scientific community. Even-though the manuscript is a bit lengthy -albeit with little if any reduction possible without a unbalanced loss in comleteness- it is also rather easy to read through. Overall a very detailed discussion of an interesting issue, presented in an inviting and comprehensice manner.

Specific comments: 1) page 2. line 30: Regarding works on extreme temperatures modelling, the authors may wish to consult: Grotjahn, R., Black, R., Leung, R. et al., 2016, North American extreme temperature events and related large scale meteoro-logical patterns: a review of statistical methods, dynamics, modeling, and trends, Clim Dyn, 46: 1151. doi:10.1007/s00382-015-2638-6 Hasan H., Fadhilah N., Radi A., and Kassim S., 2012, Modeling of Extreme Temperature Using Generalized Extreme Value (GEV) Distribution: A Case Study of Penang, World Congress on Engineering Caroni C, Panagoulia D., 2016, Non stationary modelling of extreme temperatures in a moun-tainous ares of Greece, Rev Stat, 14,1,217-228 Kharin V., Zwiers F. et al. , 2007, Changes in Temperature and Precipitation Extremes in the IPCC Ensemble of Global Coupled Model Simulations, J. of Climate, 20, 1419-1444 ...and so on.

2) page 7, line 14: The authors might wish to discuss why the haven't considered using distribution fitting statistical tests such as the Kolmogorov-Smirnov and/or the Anderson-Darling, for the assessment of the performance of the parameter estimation methods.

3) page 7, line 21: adding a reference to: Makkonen I., 2006, Plotting Positions in Extreme Value Analysis, J Applied Meteorology and Climatology, 45, 334-340 might be helpful to the less informed reader.

4) page 8, line 2: The selection of the base value for the conditional perfromance measures, namely T, to be 10 years should be better justified and supported by rele-vant references (i.e. international or national technical ordinances or standards, best practice documentation etc).

5)page 8, line 13: Since "synoptic" has a reserved meaning in meteorology you might wish to replace it with "combined plotting" or any other suitable term. throughout the manuscript.

6) page 8, lines 16-17. Please rephrase/simplify the first sentence of section 3.1: "Linear trends....model estimation."

7) page 13, line 7 : From this point on the reader has to remember that GP refers to PDS and GEV refers to AMS. For the sake of clarity, it might be advisable to replace "GP" with "GP/PDS" and 'GEV" with "GEV/AMS" in the remainder of the text.

8) page 13, lines 8-9: "(indicated by negative deviations)" only 2 out of 4 diagrams in fig 5 show negative values at low return periods.

9) page 13, line 9: "this behaviour changes in the opposite for higher returns periods" if I am reading fig. 5 correctly, this is actually true only for precipitation. This would also have an effect on the text of the Discussion (p15, l13) and Conclusions (p20, l23) sections.

10) page 13, line 12: "(ie underestimation of negative magnitude)" is not very clear- consider rephrasing as "(ie more negative values)"

11) page 19, line 32: "as well as the number of breaks set within this range" I am no tquite certain about the meaning of this phrase. Could you please clarify ?

Technical comments: 1) page 1, line 24: add "the use of" after "recommend"

2) page 2, line 32: ...for risk assessment than events...

3) Page 11 ,Table 1 caption (as well Tables 2, 3, 4 and 5): referring to the data presented as success rates even though justifiable, might also be confusing since there is no visual groupping of the columns to indicate which columns should add to a 100%. Please consider either replacing "success rates (% of records)" with "success cases" or formatting the tables in a manner allowing easy didtinction of the various groups of

data (I would expect the latter to be a bit difficult for Table 5).

4) page 15, line 20: "explication" replace with "explanation" ?

5) page 15, line 23: "same distriburion than the..." replace with "same distriburion with the..."

6) page 16, line 11: "that ony the highest events" replace with "that ony the nore extreme events"

7) page 17, line 11: "have to be balance against" replace with "have to be balanced against"

8) page 18, Figure 8: the labels on both vertical axes are missoriented (vertical instead of horizontal)

9) page 19, lines 21: " time series extremes it is referred to" replace with "time series extremes, the reader is referred to"

10) page 19, line 26: please add the abbreviations (ATSM, MTM) after the full names of the methods

11) page 21, line 6: "We emphasize the reliable" replace with "We emphasize that reliable"

12) page 21, line 8: "analyze the fit of distribution" replace with "analyze the distribution fit"

13) page 21, line 10: "and dependency introduces biases" replace with "and dependency introduce biases"

---

## Referee Comment (RC2) · D. Rosbjerg (Referee) · 18 Jan 2017

**Review of nhess-2016-373**

The comparative assessment of AMS and POT is interesting and nicely presented, but there are some basics that need to be clarified and possibly reconsidered.

Only one specific method for selecting an appropriate threshold for POT events has been applied. This choice might be crucial for the results and the conclusions, and it is not verified that the choice is optimal, although it is argued that some graphical criteria have been fulfilled. Another choice might lead to somewhat different conclusions.

The assessments are based on conditional root-mean-square deviation and conditional mean absolute error as metrics. With the condition applied (T > 10 yr) the number of observations available for calculation of the metrics is drastically reduced. For example, if the AMS sample is covering 30 yr, only the three largest observations are applicable for calculating the metrics; in a 50 yr sample only the five largest observations can be used. Taking into account that the variance of the order statistics is strongly increasing towards the upper end of the ordered sample, it is evident that the metrics become highly uncertain.

For assessment of empirical probabilities in the ordered sample the Weibull plotting position has been selected. While the choice of plotting position formula in many cases is of minor importance, it might be influential in the present case with overly weight on the upper order statistics. If F indicates a chosen probability distribution, and $y_m$ is the m'th order statistic in a sample of size N, then the Weibull plotting position stems from the fact that $E\{F(y_m)\} = m/(N+1)$. However, with M denoting the median operator, we have $F^{-1}(m/(N+1)) < M\{y_m\} < E\{y_m\}$. Thus $F^{-1}(m/(N+1))$ is relatively close to the modal value of $y_m$ (where this exists), but far from being unbiased. A more balanced and consistent choice of plotting position would be the median plotting position as, independently of F, we have $M\{F(y_m)\} = F(M\{y_m\}) \approx (m-0.3)/(N+0.4)$.

There is a basic difference between calculation of the return periods in AMS and POT, which is important for T < 10 yr. For example, a 2 yr POT event corresponds to approximately a 2.54 yr AMS event, and a 5 yr POT event to a 5.52 AMS event. It is not evident how the difference between POT and AMS return periods has been handled.

Minor errors:

Page 7, line 21: The reference Makkonen (2005) is not in the reference list.

Page 20, line 2: insert "to" after "led".

Page 21, line 4: "where" -> "were"

*Dan Rosbjerg*

---

## Author Comment (AC1) · 3 Mar 2017

**1   Response to Reviewer 1**

We would like to thank the referee for the very positive evaluation of our manuscript and the provided feedback. Please find our responses below, with referee comments in italics, and authors' responses in standard format.

**1.1 Specific Comments**

1. *page 2, line 30: Regarding works on extreme temperatures modelling, the authors may wish to consult (...).*

   Comparative studies on extreme temperature modelling are rare. Grotjahn et al. (2016) argue in favor of the POT approach for the application on large scale meteorological patterns, but the comparison is based on literature review rather than data-based analyses. We will add this information to the MS. Hasan et al. (2012), Caroni et al. (2016) and Kharin et al. (2007) appear less relevant, as they only apply a single approach (AMS) and did not compare the results to the alternative approach.

2. *page 7, line 14: The authors might wish to discuss why they haven't considered using distribution fitting statistical tests such as the Kolmogorov-Smirnov and/or the Anderson-Darling, for the assessment of the performance of the parameter estimation methods.*

   Distribution-fitting tests are primarily useful for gaining an appreciation whether a lack of fit is statistically significant, or rather an effect of sampling uncertainty, but they have little discriminative power to identify the "true" or "best" distribution to use (e.g. Stedinger, 1993). Hence, they do not provide a straightforward measure for comparing goodness-of-fit across AMS and PDS approaches. We will add a note to the text.

3. *page 7, line 21: adding a reference to Makkonen I., 2006, Plotting Positions in Extreme Value Analysis, J Applied Meteorolgy and Climatology, 45, 334–340 might be helpful to the less informed reader.*

   We will add this reference.

4. *page 8, line 2: The selection of the base value for the conditional performance measures, namely $T$, to be 10 years should be better justified and supported by*

*relevant references (i.e. international or national technical ordinances or standards, best practice documentation etc).*

Although different return periods, ranging from 2 – 100 years (and more) have been used in engineering and storm water management and no common standard about recommended return periods seems to exist, return periods of at least 5 – 10 years are often considered as a lower threshold in storm infrastructure design (e.g. GRCA, 2014; EPA, 2014). Hence, such a level appears well suited to separate expected occurrence (i.e., non-extremes) from extreme events. We will add this information to section 2.5.

5. *page 8, line 13: Since "synoptic" has a reserved meaning in meteorology you might wish to replace it with "combined plotting" or any other suitable term throughout the manuscript.*

   We will follow this suggestion and replace "synoptic" by "combined plotting".

6. *page 8, lines 16–17: Please rephrase/simplify the first sentence of section 3.1.*

   We will rephrase this sentence.

7. *page 13, line 7: From this point on the reader has to remember that GP refers to PDS and GEV refers to AMS. For the sake of clarity, it might be advisable to replace "GP" with "GP/PDS" and "GEV" with "GEV/AMS" in the remainder of the text.*

   We will replace "AMS" and "GP" with "AMS/GEV" and "PDS/GP" in sections 4 and 5 as proposed.

8. *page 13, lines 8–9: "(indicated by negative deviations)" only 2 out of 4 diagrams in Fig 5 show negative values at low return periods.*

   Thanks for pointing this out; we will modify the text accordingly.

9. *page 13, line 9: "this behaviour changes in the opposite for higher returns periods" – If I am reading fig. 5 correctly, this is actually true only for precipitation.*

*This would also have an effect on the text of the Discussion (p15, l13) and Con-clusions (p20, l23) sections.*

We will rephrase the respective sections accordingly.

10. *page 13, line 12: "(ie underestimation of negative magnitude)" is not very clear – consider rephrasing as "(ie more negative values)"*

    We will rephrase the section containing this sentence.

11. *page 19, line 32: "as well as the number of breaks set within this range" I am not quite certain about the meaning of this phrase. Could you please clarify?*

    Both methods employ a sequence of thresholds that are generated based on a specified range and resolution of values. We will revise the entire paragraph (also with reference to the comment of Reviewer #2) to improve readability.

**1.2 Technical Comments**

All technical comments have been implemented as proposed by the reviewer.

**1.3 References**

- GRCA: Technical and engineering guidelines for stormwater management submissions, Ganaraska Region Conservation Authority, Port Hope, Ontario, available at: http://www.grca.on.ca/Guidelines_for_swm_submissions-_FINAL.pdf (Accessed 24 February 2017), 2014.

- EPA: Addressing green infrastructure design challenges in the Pittsburgh region – Abundant and frequent rainfall, United States Environmental Protection Agency, Pittsburgh, Pennsylvania, available at: http://www.3riverswetweather.org/sites/default/files/Rainfall%20white%20paper.pdf (Accessed 24 February 2017), 2014.

- Stedinger, J. R., Vogel, R. M. and Foufoula-Georgiou, E.: Frequency analysis of extreme events, Chapter 18 in Handbook of Hydrology, edited by DR Maidment, McGraw-Hill., 1993.

**2   Response to Reviewer 2**

We also would like to thank Dan Rosbjerg for his useful feedback. Our response is given below, with referee comments in italics, and our responses in standard format.

1. *Only one specific method for selecting an appropriate threshold for POT events has been applied. This choice might be crucial for the results and the conclusions, and it is not verified that the choice is optimal, although it is argued that some graphical criteria have been fulfilled. Another choice might lead to somewhat different conclusions.*

   We agree that different thresholds might lead to somewhat different conclusions. However, appropriate threshold choice is one of the most discussed issues related to the threshold excess approach (e.g. Scarrot and MacDonald, 2012). We have discussed this issue in the discussion section "Secondly, threshold selection in the threshold excess method is a legitimate subject for debate..."
   In this study, 100 time series (i.e. 25 stations with 4 meteorological indicators each) have been analyzed. In order be able to perform a standardized and reproducible threshold selection for this large number of time series, we decided to use some sort of supervised automated threshold selection method.
   In this respect we have tested several automated threshold selection methods, to be precise ATSM (automated threshold selection method) by Thompson et al. (2009) and MTM (multiple threshold selection method) by Deidda (2010). However, both methods yielded dissatisfying and inconsistent thresholds. Threshold values of similarly distributed time series obtained by ATSM varied considerably,

and parameter estimates of MTM were depending on range and resolution of the thresholds considered. While certain patterns of convergence were found based on sensitivity analysis, we argue that these procedures somehow replace the threshold selection problem with that of selecting an appropriate range and an appropriate number of breaks (c.f. response #11 to reviewer 1).

Having tested several options, the square-root-rule criterion by Ferreira et al. (2003) – which has been used in various other studies as well (Scarrot and MacDonald, 2012) – has been employed. The results have been double-checked by means of diagnostic plots for threshold selection (mean residual life plot, parameter stability plots), which are the sole basis for threshold selection in many other studies (c.f. Coles, 2001; Della-Marta et al., 2009; Scarrott and MacDonald, 2012). Results show that the thresholds derived in this way provide reasonable results (c.f. also Figure 8). We will revise and enhance the discussion section accordingly (also, with ref. to Reviewer 1) in order to clarify the issues pointed out by the reviewer.

2. *The assessments are based on conditional root-mean-square deviation and conditional mean absolute error as metrics. With the condition applied ($T > 10\ yr$) the number of observations available for calculation of the metrics is drastically reduced. For example, if the AMS sample is covering 30 yr, only the three largest observations are applicable for calculating the metrics; in a 50 yr sample only the five largest observations can be used. Taking into account that the variance of the order statistics is strongly increasing towards the upper end of the ordered sample, it is evident that the metrics become highly uncertain.*

Note that we are not solely analyzing conditional errors for the desired extremes ($T > 10\ yr$), but also the overall G.O.F. We think this specific assessment of the desired extremes is important, as the overall G.O.F is mainly representing the non-extreme part which is usually of little relevance. However, we agree that the proposed metrics are of limited robustness, especially if time series are short and

the condition is selected for high return periods.

Depending on the length of the time series available, the value for $x$ should be chosen accordingly. In our study, most of our time series date back to the period between the world wars, or even further back as early as 1895. Choosing $T > 10\ yr$ seems feasible in these cases.

We will add this point to the discussion.

3. *For assessment of empirical probabilities in the ordered sample the Weibull plotting position has been selected. While the choice of plotting position formula in many cases is of minor importance, it might be influential in the present case with overly weight on the upper order statistics. If F indicates a chosen probability distribution, and $y_m$ is the m'th order statistic in a sample of size N, then the Weibull plotting position stems from the fact that $E[F(y_m)] = m/(N+1)$. However, with $M$ denoting the median operator, we have $F-1(m/(N+1)) < M[y_m] < E[y_m]$. Thus $F-1(m/(N+1))$ is relatively close to the modal value of $y_m$ (where this exists), but far from being unbiased. A more balanced and consistent choice of plotting position would be the median plotting position as, independently of $F$, we have $M[F(y_m)] = F(M[y_m]) \approx (m-0.3)/(N+0.4)$.*

The choice of a plotting position (PP) was mainly important when distribution parameters were estimated graphically from a probability paper. In this paper, according to common standard, parameters are estimated using analytical equations (e.g. based on L-moments method or maximum likelihood method) which do not depend on the choice of plotting positions. The choice of a specific plotting position is therefore of minor relevance for our paper.

Only the conditional performance measure depends, to some extent, on the chosen PP, as it uses a quantile estimate related to $T = 10$ years to select extreme values. To assess the sensitivity of the measure to the choice of the PP, we computed G.O.F. results with two alternative measures (i.e. based on Beard (median) and Gringorton plotting position) for the GEV case. We found that differences in

CRMSE$_{10}$ and CMAE$_{10}$ values are only minor. As far as Weibull PP and median PP are concerned, mean absolute deviations in CRMSE$_{10}$ values are around $0.09 - 0.17\,°C$ for the different temperature indices and $1.15\,mm$ for precipitation. Summary statistics of absolute CRMSE$_{10}$ deviations between Weibull PP and median PP (including parameter estimation based on both MLE and LMOM for each of the four meteorological indicators) are presented below:

```
> sapply(delta_abs_crmse, summary)
          dt        precip   tmax        tmin
 Min.     0.008714  0.08059  0.0005396   0.02376
 1st Qu.  0.060140  1.12000  0.0513100   0.13130
 Median   0.084670  1.69500  0.0863700   0.15180
 Mean     0.093220  1.68000  0.0960000   0.16620
 3rd Qu.  0.100400  2.15100  0.1263000   0.17920
 Max.     0.218600  3.69600  0.2328000   0.35560

> sapply(delta_abs_cmae, summary)
          dt       precip   tmax       tmin
 Min.     0.00146  0.01179  0.002527   0.0009262
 1st Qu.  0.03299  0.53900  0.033770   0.0968800
 Median   0.06176  1.16600  0.060990   0.1352000
 Mean     0.07185  1.22500  0.074560   0.1432000
 3rd Qu.  0.07904  1.67300  0.086700   0.1693000
 Max.     0.20000  2.96400  0.218500   0.3472000
```

When using median PP instead of Weibull PP, results in terms of best fitting estimation method change in 16 % of all cases. However, in these cases, effects on results in terms of return level estimates are only minor, since – except for one case – changes occurred in cases where both estimation methods yield very similar parameter estimates. Results of the differences in return levels based on
Beard and Weibull PP are presented below:

```
**return levels w/ Beard PP - return levels w/ Weibull PP**
> delta_rl
$dt
   rl10  rl20  rl50 rl100
1 -0.09 -0.15 -0.21 -0.26
2 -0.03 -0.02  0.01  0.03
3  0.02 -0.01 -0.07 -0.16
4 -0.04 -0.04 -0.03 -0.01

$precip
   rl10  rl20  rl50  rl100
1 -0.44 -2.12 -5.97 -10.47
2 -0.45 -0.73 -1.16  -1.52
3  0.09 -0.39 -1.46  -2.64
4 -0.04 -0.05 -0.14  -0.26
5 -0.67 -0.80 -0.91  -0.95

$tmax
   rl10  rl20  rl50 rl100
1 -0.12 -0.21 -0.35 -0.46

$tmin
   rl10  rl20 rl50 rl100
1  0.22  0.50 0.92  1.23
2 -0.07 -0.04 0.05  0.15
3  0.34  0.73 1.25  1.61
4  0.33  0.51 0.76  0.94
5  0.13  0.57 1.26  1.85
```
```
6  0.17  0.40 0.70  0.91
```

Additional plots showing a graphical comparison between Weibull, Beard and Gringorten PP for all four parameters and for all 25 stations each are attached in the supplement.

In addition, our decision to use Weibull PP is based on the fact that we wanted to find a common ground regarding PP. Weibull PP is the one most commonly used in extreme value analysis and has been applied in most reference works in this area (e.g. Coles, 2001). We basically followed the argumentation of Lasse Makkonen (2005, 2008, 2013), who argues that the Weibull plotting position is the most suitable plotting position, independent of the underlying distribution $f(x)$.
For further information, see the following publications and the cited references (also including opposing views) therein:

- Makkonen, L. (2005): Plotting Positions in Extreme Value Analysis. Journal of Applied Meteorology and Climatology, 45: 334–340. doi:10.1175/JAM2349.1. Available at: http://journals.ametsoc.org/doi/pdf/10.1175/JAM2349.1.

- Makkonen, L. (2008): Bringing Closure to the Plotting Position Controversy. Communications in Statistics – Theory and Methods, 37: 460–467. doi:10.1080/03610920701653094.

- Makkonen, L.; Pajari, M. & Tikanmäki M. (2013): Closure to "Problems in the extreme value analysis" (Struct. Safety 2008:30:405–419). Structural Safety, 40: 65–67. doi:10.1016/j.strusafe.2012.09.007.

4. *There is a basic difference between calculation of the return periods in AMS and POT, which is important for $T < 10\ yr$. For example, a 2 yr POT event corresponds to approximately a 2.54 yr AMS event, and a 5 yr POT event to a 5.52 AMS event. It is not evident how the difference between POT and AMS*

*return periods has been handled.*

Thanks for pointing this out; it is perfectly right that AMS return periods ($T$) and POT return periods ($T_*$) are not the same, they are rather related in the form of

$$\frac{1}{T} = 1 - e^{\frac{-1}{T_*}}.$$

This inequality was considered by converting AMS return periods to PDS return periods (in order to avoid underestimating the probability of occurrence). However, it turned out that this has not been done in Fig. 5. We have clarified this issue in the methodology section and made corrections to Fig. 5. References to Langbein (1949), Rosbjerg (1977) and Madsen et al. (1997) have been added.

Please also note the supplement to this comment:
http://www.nat-hazards-earth-syst-sci-discuss.net/nhess-2016-373/nhess-2016-373-AC1-supplement.zip